# Hypoxia Enhances the Expression of RNASET2 in Human Monocyte-Derived Dendritic Cells: Role of PI3K/AKT Pathway

**DOI:** 10.3390/ijms22147564

**Published:** 2021-07-15

**Authors:** Sara Monaci, Federica Coppola, Gaia Giuntini, Rossella Roncoroni, Francesco Acquati, Silvano Sozzani, Fabio Carraro, Antonella Naldini

**Affiliations:** 1Cellular and Molecular Physiology Unit, Department of Molecular and Developmental Medicine, University of Siena, 53100 Siena, Italy; sara.monaci@student.unisi.it (S.M.); federica.coppola@student.unisi.it (F.C.); gaia.giuntini@student.unisi.it (G.G.); 2Department of Biotechnologies and Life Sciences, University of Insubria, 21100 Varese, Italy; rroncoroni@studenti.uninsubria.it (R.R.); Francesco.Acquati@uninsubria.it (F.A.); 3Laboratory Affiliated to Istituto Pasteur Italia-Fondazione Cenci Bolognetti, Department of Molecular Medicine, Sapienza University of Rome, 00185 Rome, Italy; silvano.sozzani@uniroma1.it; 4IRCCS Neuromed, 86077 Pozzilli, Italy; 5Department of Medical Biotechnologies, University of Siena, 53100 Siena, Italy; Fabio.Carraro@unisi.it

**Keywords:** hypoxia, dendritic cell, RNASET2

## Abstract

Hypoxia is a key component of the tumor microenvironment (TME) and promotes not only tumor growth and metastasis, but also negatively affects infiltrating immune cells by impairing host immunity. Dendritic cells (DCs) are the most potent antigen-presenting cells and their biology is weakened in the TME in many ways, including the modulation of their viability. RNASET2 belongs to the T2 family of extracellular ribonucleases and, besides its nuclease activity, it exerts many additional functions. Indeed, RNASET2 is involved in several human pathologies, including cancer, and it is functionally relevant in the TME. RNASET2 functions are not restricted to cancer cells and its expression could be relevant also in other cell types which are important players in the TME, including DCs. Therefore, this study aimed to unravel the effect of hypoxia (2% O_2_) on the expression of RNASET2 in DCs. Here, we showed that hypoxia enhanced the expression and secretion of RNASET2 in human monocyte-derived DCs. This paralleled the HIF-1α accumulation and HIF-dependent and -independent signaling, which are associated with DCs’ survival/autophagy/apoptosis. RNASET2 expression, under hypoxia, was regulated by the PI3K/AKT pathway and was almost completely abolished by TLR4 ligand, LPS. Taken together, these results highlight how hypoxia- dependent and -independent pathways shape RNASET2 expression in DCs, with new perspectives on its implication for TME and, therefore, in anti-tumor immunity.

## 1. Introduction

The tumor microenvironment (TME) has an undeniable influence on cancer progression by affecting tumor growth and on the ability of stromal and immune cells to orchestrate immune responses locally [1]. Hypoxia is a key component of the TME and a severe intratumoral hypoxia is associated with increased risk of mortality [2]. Indeed, hypoxia promotes not only tumor growth and metastasis, but it negatively affects infiltrating immune cells by impairing host immunity [3].

Dendritic cells (DCs) are the most potent antigen-presenting cells [4] and during their lifespan they are often exposed to hypoxia [5,6].

Although DCs constitute a rare immune cell population within tumors, these cells are crucial for initiation and regulation of the immune responses in a microenvironment, like the TME, which is characterized by a low oxygen tension [7,8]. In this context, tumors impair DC biology in many ways, including the modulation of DC differentiation, and the modification of their metabolism by decreasing the availability of nutrients and oxygen, and, therefore, by compromising DC viability [3]. Previous reports have shown that DCs are affected by hypoxia in terms of cell survival, differentiation, activation, and migration [9,10]. In addition, we have previously demonstrated that a prolonged exposure to hypoxia resulted in a pro-apoptotic program in immature DCs, along with PI3K/AKT inhibition [11].

The PI3K/AKT pathway plays a key role in DC survival [12]. This pathway is tightly related to the hypoxia-inducible factor (HIF) which is the master regulator of the hypoxia responses [13]. HIF is composed by an oxygen-regulated α subunit and a constitutively expressed β subunit. In the presence of oxygen, HIF-1α is prolyl-hydroxylated and, thereafter, it interacts with the von Hippel Lindau protein, leading to ubiquitination and rapid destruction in proteasomes. In hypoxia, HIF-1α evades proteasomal degradation. This results in the binding of active HIF heterodimers with the hypoxic-responsive elements (HREs) present in the promoter of the target genes, including the C-X-C chemokine receptor type 4 (CXCR4), the pro-angiogenic vascular endothelial growth factor (VEGF-A), and the BCL-2 Interacting Protein 3 (BNIP3). The latter belongs to the BCL-2 family of proteins, and, along with other BCL-2 family members, such as the induced myeloid leukemia cell differentiation protein Mcl-1, is involved in DC survival, autophagy, and apoptosis [14,15,16].

RNASET2 belongs to the T2 family ribonucleases, which are highly conserved, since they are found in the genomes of protozoans, plants, bacteria, animals, and viruses [17]. Besides its nuclease activity, RNASET2 exerts many additional functions and it has been involved in several human pathologies, including inflammation and cancer [18]. RNASET2 has been associated with anti-tumor activities, since its overexpression is a good prognostic index in several neoplastic diseases [19]. Indeed, its overexpression inhibits the clonogenicity of ovarian cancer cells in vitro and suppresses tumorigenesis and metastatic potential in vivo [20,21]. Besides its intracellular expression, RNASET2 is also secreted and recent experimental data from several groups have indicated that human RNASET2 belongs to the stress-response gene family [22]. The role of RNASET2 is not restricted to cancer cells and could be relevant also in other cell types which are important players in the TME. In fact, RNASET2 has been recently associated with immune cell functions, highlighting its potential involvement in innate and, so far, anti-tumor immunity [18,23,24,25]. Thus, the modulation of RNASET2 expression and secretion in DCs may be functionally relevant in the TME.

In this study, and for the first time, we report that hypoxia enhances the expression and secretion of RNASET2 in human monocyte-derived DCs. This was paralleled by the HIF-1α accumulation and HIF-dependent and -independent signaling, and with a decreased DCs survival. Interestingly, we found out that RNASET2 overexpression, under hypoxia, was enhanced by pharmacological inhibition of PI3K/AKT. Furthermore, treatment of hypoxic DCs with the TLR4 ligand LPS resulted in the almost complete abolishment of RNASET2 expression, which was reversed by PI3K/AKT inhibition.

## 2. Results

### 2.1. Hypoxia Enhances RNASET2 Expression Along with HIF-1α Dependent and Independent Signaling in Human Monocyte-Derived DCs

To determine the effects of hypoxia on RNASET2 expression, DCs were exposed to normoxia and hypoxia for 24 h. The hypoxic condition that was used in the study (2% O_2,_ equivalent to 14 mmHg), corresponding to an average of pO_2_ tension present in the TME [8], was able to induce a significant increase in HIF-1α accumulation (Figure 1A). The effects of hypoxia on DCs were not restricted to HIF-1 signaling. Indeed, hypoxia enhanced the expression of the signal transducer and activator of transcription (STAT)3α, which is typically associated with DC inhibition within the TME [26,27]. In addition, we determined whether our hypoxic experimental conditions resulted in the overexpression of genes, which are transcriptionally regulated by HIF-1α. These include the BCL-2 family member BNIP3, the pro-angiogenic growth factor VEGF-A and the chemokine receptor CXCR4. BNPI3 induces loss of mitochondrial membrane potential, which may promote DC cell death, whereas VEGF-A is commonly referred as a promoter of tumor angiogenesis and CXCR4-CXCL12 signaling has been associated with the polarization towards an immune-suppressive microenvironment [28]. Having established that our experimental conditions were compatible with a hypoxic response/signature in DCs [11], we next determined whether hypoxia could affect RNASET2 expression. In this regard, when DCs were exposed to hypoxia they showed a significant and sustained enhancement of RNASET2 mRNA expression, as established by RT-qPCR (Figure 1B). Moreover, we observed that hypoxia significantly increased RNASET2 protein levels, when compared to the normoxic control. Furthermore, the increased RNASET2 expression was not only observed at intracellular level. Indeed, a 48 h exposure of DCs to hypoxia also caused a significant increase in RNASET2 secretion, as determined by ELISA, indicating its potential effects on the surrounding cells within the TME.

The hypoxia-induced intracellular and extracellular protein levels of RNASET2 were paralleled by a decreased DC viability, which was assessed with ethidium bromide/calcein staining and fluorimetric analysis. As shown in Figure 1C, the percentage of live DCs was significantly lower after a 48 h exposure to hypoxia when compared with normoxic controls. In addition, exposure to hypoxia was accompanied by a decreased phosphorylation of AKT and mTOR. The enhanced DC cell death program by hypoxia was further confirmed by the decrease in Mcl-1 that is a member of the BCL-2 protein family, with specific antiapoptotic effects in DCs [16,29].

### 2.2. Modulation of Hypoxia-Induced RNASET2 Expression by PI3K/AKT Pathway and TLR4 Ligand LPS

The above results suggest that PI3K/AKT may be associated with RNASET2 expression in hypoxic DCs. The involvement of the PI3K/AKT pathway in the regulation of RNASET2 was established by a series of experiments, where DCs were exposed under hypoxia for 24 h employing, in the last 6 h, the panPI3K irreversible inhibitor Wortmannin [30]. As shown in Figure 2A, treatment of DCs with Wortmannin under hypoxia significantly enhanced the expression of RNASET2. The effectiveness of Wortmannin under hypoxia, was validated by a significant inhibition of AKT and mTOR phosphorylation, as well as by a reduced expression of the anti-apoptotic protein Mcl-1, which is specifically related to cell survival in immature DCs [29].

To further test the mechanism by which hypoxia affects RNASET2 expression in DCs, we performed additional experiments employing an activator of PI3K/AKT signaling in DCs, the TLR4 ligand, LPS. As shown in Figure 2B, a 24 h exposure to hypoxia in the presence of LPS resulted in an increased phosphorylation of AKT and mTOR, when compared to hypoxic controls. This reduction was accompanied by a significant increase in the antiapoptotic protein Mcl-1, confirming a previous study showing that PI3K/AKT activation by LPS protects DCs from apoptosis under hypoxia [11]. More interestingly, LPS treatment caused a significant reduction in RNASET2 expression at both mRNA and protein levels (Figure 2C). As we postulated that RNASET2 expression was regulated by the PI3K/AKT pathway, we treated DCs with LPS also in the presence of the PI3K inhibitor, Wortmannin. In this case, the inhibitor caused a significant enhancement of RNASET2 protein expression in LPS-treated DCs under hypoxia. Once again, these results suggest an important implication of the PI3K/AKT pathway in RNASET2 modulation, thus indicating a novel and additional modality by which RNASET2 may be regulated in DCs upon their exposure to a hypoxic condition, like in the TME.

## 3. Discussion

Here, we report for the first time that hypoxia enhances the expression of RNASET2 in human DCs and that PI3K/AKT and TLR4-activation by LPS modulate such expression.

Previous reports have shown that RNASET2 is overexpressed in ovarian cancer lines in response to several stress conditions, including hypoxia [19]. The same report showed that in RNASET2-silenced OVCAR3 ovarian cancer cell line, under hypoxic conditions, the proliferation rate was significantly higher when compared to controls [19]. In addition, its overexpression inhibits the clonogenicity of ovarian cancer cells in vitro [20,21], indicating that RNASET2 may be implicated in the inhibition of tumor cell proliferation and survival.

Of note, the fact that intracellular RNASET2 may control cell proliferation/survival is not restricted to tumor cells. Indeed, a previous study has shown that stress-induced RNASET2 overexpression inhibits cell proliferation and mediates apoptosis in human melanocytes [22]. In keeping with these data, we found out in our study that RNASET2 overexpression in hypoxic DCs was accompanied by a decreased cell viability, along with downregulation of anti-apoptotic proteins belonging to the BCL-2 family [29]. One of them, Mcl-1, is particularly important for DC survival and differentiation [15]. In addition, we observed that hypoxic DCs concomitantly showed lower phosphorylation levels of AKT and mTOR, and both belong to signaling pathways which are crucial for DC survival [31,32]. More importantly, we showed that the induction of RNASET2 under hypoxia was further increased by PI3K/AKT inhibition, supporting the involvement of pro-survival signaling pathway in the modulation of RNASET2 expression in hypoxic DCs.

The fact that hypoxia induces a massive increase in expression and secretion of RNASET2 may be related to a previous observation where in ovarian cancer cell lines, analysis of a region immediately upstream of the first exon of the RNASET2 gene and the whole first intron showed two putative HIF-1 binding sites, suggesting a possible role for one or more of these elements in the observed hypoxia-induced upregulation of RNASET2 expression [19]. In the present study, hypoxic DCs expressed high levels of RNASET2 mRNA, and protein levels, which paralleled the accumulation of HIF-1α and the expression of genes which are transcriptionally regulated by HIF-1α. One of them is VEGF-A which, in the TME, contributes to the inhibition of DC functions. The other gene that is regulated by HIF is BNIP3 which plays an important role in autophagy/apoptosis. Thus, RNASET2 may be one of the genes that could be relevant for modulating DC survival under hypoxia as well.

In addition, the observed increase in RNASET2 secretion in DCs’ response to hypoxia further supports the hypothesis of its putative role as an “alarmin”, by activating a “danger-response” program, with important implications in the TME [33]. Alarmins are endogenous molecules which share some common features, including their passive release from necrotic cells or active secretion from cells of the innate immune system, that promote adaptive immunity [34]. Thus, the observed secretion of RNASET2 by hypoxic DCs suggests the execution of extracellular roles which could be relevant in the TME to coordinate an immune response.

The impact of RNASET2 in DCs functions was suggested by a previous study where *Clonorchis sinensis*-derived RNASET2 could significantly suppress the expression of LPS-induced DCs maturation markers in murine bone marrow derived DCs [35]. Our study extends this observation to endogenous RNASET2 and characterize the PI3K/AKT as the major pathway involved in LPS-induced downregulation in DCs.

The cross talk between TLRs and RNASET2 in immune sensing has been evidenced by two recent reports, showing that TLR8 is a sensor of RNASET2 degradation products [24,25]. The latter one clearly indicates the relevance of lysosomal RNASET2 activity functions upstream of TLR8 in monocytic cell lines. However, the authors suggest that RNASET2’s pro-immunogenic role upstream of TLR8 may be in a tight equilibrium with its potential anti-inflammatory role as an RNA-degrading enzyme. In our study, we have shown that RNASET2 expression may be modulated by LPS, thus indicating a potential function downstream of TLR4. It is tempting to speculate that in DCs the pro-inflammatory effects, which are physiologically activated by TLR4 activation, may be followed by a strong downregulation of intracellular RNASET2 expression and, so far, a reduced anti-inflammatory effect. Still, regarding LPS, it is widely accepted that it induces a terminal differentiation in DCs [36]. Thus, RNASET2 expression may be inversely associated with DC maturation and its intracellular decrease, as we observed upon LPS stimulation, may ensue a pro-survival effect in DCs, therefore protecting them against the hypoxic stress. The inhibition of DCs’ viability by TME is not restricted to the hypoxic microenvironment. Tumor-derived factors, such as VEGF-A, PGE2, and ATP, keep infiltrating DCs at an immature state that is not functional for the activation of T cell response [10]. Indeed, one of the characteristics of the TME is the accumulation of immature DCs that are not able to finally differentiate. This accumulation generates tolerogenic signals resulting in a non-execution of a proper anti-tumor immune response. RNASET2 upregulation, together with the inhibition of phAKT, phmTOR, and Mcl-1 might be part of an “editing” process that leads to the elimination of pro-tumor immature DCs. This process will abort in the case DCs sense activation signals, such as LPS, leading to maturation and anti-tumor immunity.

Our hypotheses, based mainly on observations and correlations, need to be further corroborated by more in-depth experiments concerning the relation between RNASET2 expression and the DC maturation profile in hypoxic condition. Hypoxia itself modulates the maturation of DCs, resulting in a reduced expression of markers of their maturation in the presence of LPS [37]. In this regard, specific experiments are currently ongoing. Preliminary data confirm the reduction in some maturation markers in hypoxic DCs and this reduction appears to correlate with the increased expression of RNASET2 (data not shown).

In conclusion, we have established that a prolonged hypoxia, along with the inhibition of DCs’ survival, induces the expression of RNASET2 and that this effect is negatively regulated by the PI3K/AKT pathway and LPS. Our study, albeit preliminary, opens new perspectives on the role of RNASET2 in a hypoxic microenvironment as TME and so in anti-tumor immunity.

## 4. Materials and Methods

### 4.1. Reagents

RPMI 1640, fetal bovine serum (FBS), penicillin/streptomycin, L-Glutamine were purchased from Euroclone, Devon, UK. Fycoll was purchased from Cederlane Labs and Percoll from Amersham Bioscience, Pittsburgh, PA, USA. Recombinant human granulocyte macrophage colony stimulating factor (GM-CSF) and interleukin-13 (IL-13) were purchased from ProSpec TechnoGene, East Brunswick, NJ, USA. All reagents contained <0.125 endotoxin units/mL, as checked by the Limulus Amebocyte Lysate assay (Cambrex, East Rutherford, NJ, USA). LPS from Escherichia coli strain 055:B5 was obtained from Sigma–Aldrich, Milano, Italy. Wortmannin was purchased from Tocris Biosciences, Bristol, UK.

### 4.2. Human Monocyte-Derived DC Preparation and Culture Conditions

Human monocyte-derived DCs were generated as previously described [11]. The study was reviewed and approved by Ethical Committee of Azienda Ospedaliera Universitaria Senese and University of Siena (CAVSE 17022020). The participants provided their written informed consent.

Briefly, highly enriched blood monocytes (>95% CD14) were obtained from anonymous buffy coats (South-East Tuscany Blood Establishment, AOUS, Siena, Italy) by Fycoll and Percoll gradient centrifugations. Monocytes were differentiated into DCs (>90% CD1a and <5% CD14) upon 6 days culture (in RPMI 1640, supplemented with 10% FBS) with 50 ng/mL GM-CSF and 20 ng/mL IL-13, as previously reported [11].

DCs were cultured under either normoxia (atmospheric pO_2_ levels: 21% O_2_, 5% CO_2_ and 74% N_2_ corresponding to a pO_2_ ~ 140 mmHg) or hypoxia (2% O_2_, 5% CO_2_ and 94% N_2_, corresponding to a pO_2_ ~ 14 mmHg) by the workstation InVivo O_2_ 400 (Ruskinn, Pencoed, UK) as previously described [11]. In some experiments, LPS (100 ng/mL), was added and the cultures were carried out for 24 h.

Where indicated, Wortmannin (5 µM) was added directly to the culture medium 6 h before the end of treatment. At the indicated times, cells were harvested for further analysis, as described below.

### 4.3. Western Blot

DCs were harvested and lysed in 40 µL of RIPA buffer (Cell Signaling Technologies, Danvers, MA, USA) containing a cocktail of protease inhibitors (Sigma-Aldrich, St. Louis, MO, USA). Then, equal amounts of total proteins were loaded onto SDS-PAGE gel and blotted onto a nitrocellulose membrane (BIO-RAD, Hercules, CA, USA) and blocked in TBS supplemented with 0.1% Tween and 5% nonfat dry milk for 1 h. The following primary antibodies were used as listed: HIF-1α (BD Biosciences, San Jose, CA, USA, 1:200 Cat.n° 610958), STAT3α (Cell Signaling Technologies, Danvers, MA, USA, 1:1000 Cat.n° 9139), RNASET2 (rabbit polyclonal antibody raised against recombinant RNASET2 protein, (kindly provided by Francesco Acquati) phAKT (Cell Signaling Technologies, Danvers, MA, USA, 1:1000 Cat.n°4058), phmTOR (Cell Signaling Technologies, Danvers, MA, USA, 1:1000 Cat.n° 2971), Mcl-1 (Cell Signaling Technologies, Danvers, MA, USA, 1:1000 Cat.n° 94296), and β-actin (Sigma-Aldrich, 1:50,000 Cat.n° A3854). Anti-mouse IgG HRP (Cell Signaling Technologies, Danvers, MA, USA, 1:2000 Cat.n° 7076) and anti-rabbit IgG-HRP (Cell Signaling Technologies, Danvers, MA, USA, 1:2000 Cat.n° 7074) were used as secondary antibodies (Cell Signaling Technologies, Danvers, MA, USA). Detection of images was completed using ChemiDoc™ MP System (BIO-RAD, Hercules, CA, USA). Blots were quantified using Image Lab software (BIO-RAD, Hercules, CA, USA).

### 4.4. RNA Isolation Extraction and RT-qPCR

Total RNA was isolated using EuroGOLD™Trifast reagent (Euroclone, Devon, UK) and reverse-transcribed with iScript™cDNA Synthesis Kit (Bio-Rad Laboratories, Hercules, CA, USA) according to the manufacturer’s instructions. RT-qPCR analysis was performed using SsoAdvanced™ Universal SYBR^®^ Green Supermix (Bio-Rad Laboratories, Hercules, CA, USA) and mRNA levels of BNIP3, VEGF-A, CXCR4, and RNASET2 were determined by MiniOPTICON™ System (Bio-Rad Laboratories). Data were quantitatively analyzed on an iQ5™ Optical System Software (Bio-Rad Laboratories) by using the 2^−ΔΔCT^ method and β—actin was used as housekeeping gene.

The sequence of the RNASET2 primers used for RT-qPCR is the following:

RNASET2 fw: 5′-CGTAATTCACTCGTTTCCCAATC-3′

RNASET2 rev: 5′-CCCATGCTTTTCCCACTCAT-3′

### 4.5. ELISA

Cell supernatants were collected and used for the detection of RNASET2 by a double-antibody sandwich ELISA, according to manufacturer’s instructions (BIOMATIK Ontario, Kitchener, ON, Canada, Cat. n°EKU07128). Briefly, standards and samples properly diluted were incubated onto 96 microtiter plate wells for 2 h at 37 °C. Then, a biotinylated antibody specific to RNASET2 was added for 1 h at 37 °C. After washing, avidin conjugated to horseradish peroxidase was added to each well for 30 min at 37 °C and the development was performed using TMB substrate. The TMB reaction was stopped with sulphuric acid solution and absorbance was measured at 450 nm with MULTISKAN (ThermoFisher Scientific, San Jose, CA, USA). To subtract high background signals, a reference measurement at 650 nm was performed.

### 4.6. Cell Death/Viability Assay

Percentage of live DC was assessed using a plasma membrane integrity assay (LIVE/DEAD^®^ Viability/Cytotoxicity Assay, Molecular Probes, Eugene, OR) as previously described [11]. Live DC were stained with Calcein AM, that was hydrolyzed in live cells producing an intense uniform green fluorescence (ex/em ~485 nm/~520 nm). Instead, ethidium homodimer—1 was used to detect dead cells since it only enters cells with damaged membranes and binds nucleic acids producing a bright red fluorescence (ex/em ~530 nm/~645 nm). Fluorescence was measured using a microplate reader (FLUOstar Optima, BMG Labtech, Durham, NC, USA). Live cell percentages were calculated as follows:
[live cell number/total cell number (live plus dead cells)] × 100.


### 4.7. Statistical Analysis

The data are presented as the mean ± SEM of at least 3 independent experiments. Statistical analyses were performed with Graph-Pad Prism (San Diego, CA, USA). Analysis of variance (ANOVA) and an unpaired two-tailed Student’s *t*-test were used to test for significant numerical differences among the group. Difference of *p* ≤ 0.05 was considered to be statistically significant (* *p* ≤ 0.05).

## 5. Conclusions

In conclusion, we have established that a prolonged hypoxia, along with the inhibition of DCs’ survival, induces the expression of RNASET2 and that this effect is negatively regulated by the PI3K/AKT pathway and LPS. Our study, albeit preliminary, opens new perspectives on the role of RNASET2 in a hypoxic microenvironment as TME and, therefore, in anti-tumor immunity.

## Figures and Tables

**Figure 1 ijms-22-07564-f001:**
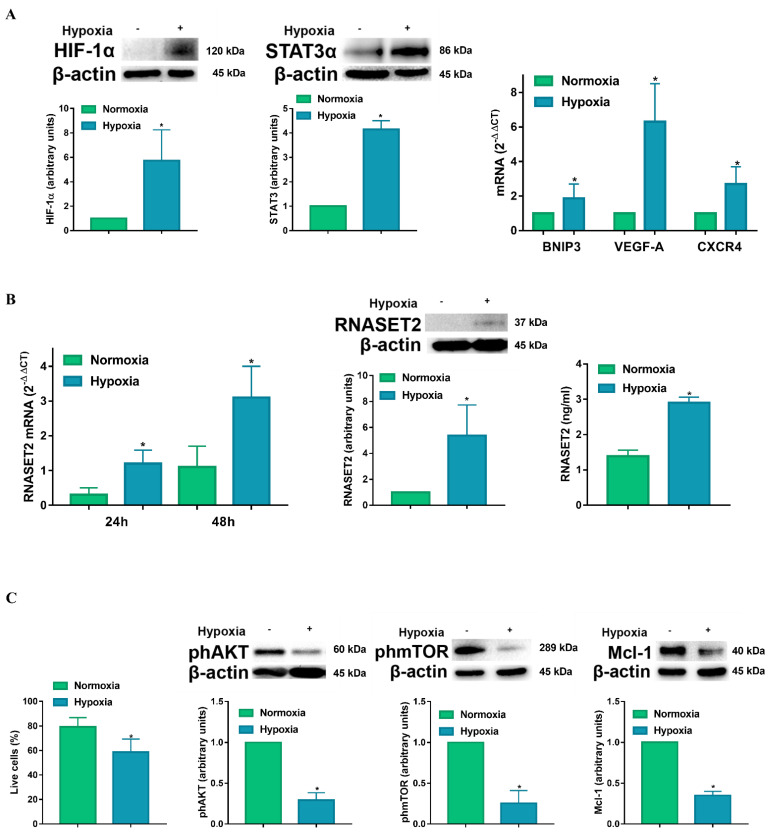
(**A**) HIF-1α and STAT3α protein levels as determined by Western blotting and BNIP3; VEGF-A and CXCR4 mRNAs as determined by RT-qPCR analysis after 24 h exposure to normoxia and hypoxia. (**B**) RNASET2 mRNA after 24 and 48 h exposure to normoxia and hypoxia as determined by RT-qPCR, RNASET2 protein levels, as detected by Western Blotting (after 24 h), and secretion as measured by ELISA assay (after 48 h). (**C**) Live cell percentage after 48 h exposure to normoxia and hypoxia, as determined by LIVE/DEAD assay, phAKT, phmTOR, and Mcl-1 protein levels after 24 h exposure to normoxia and hypoxia, as determined by Western blotting. All blots shown are representative of at least three independent experiments and β-actin was used as loading control. β-actin was used as a housekeeping gene for RT-qPCR analysis. * indicates statistically significant differences (*p* ≤ 0.05; *n* = 3).

**Figure 2 ijms-22-07564-f002:**
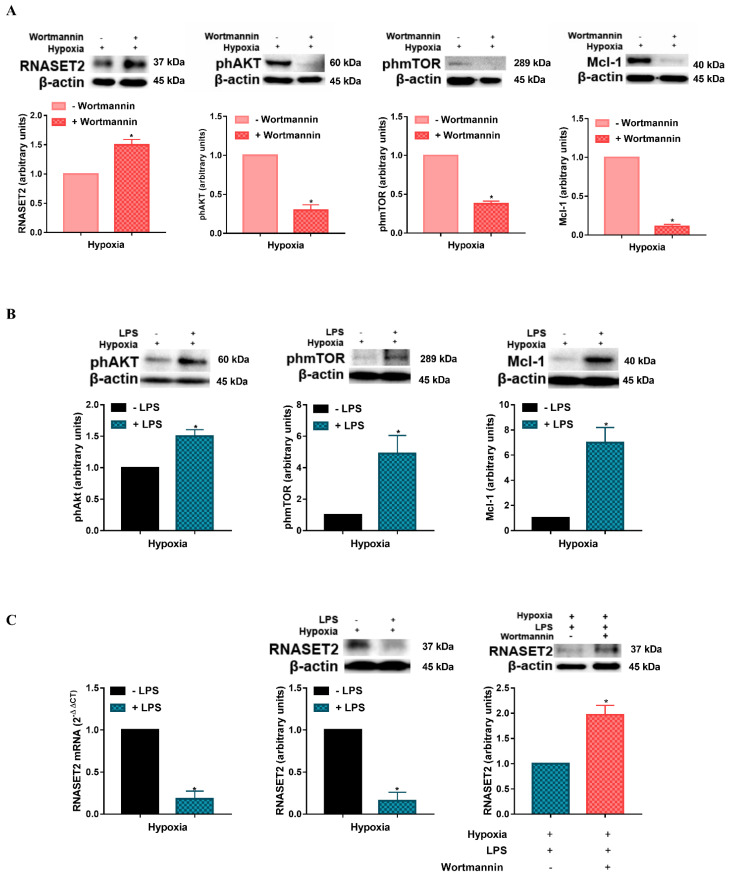
(**A**) RNASET2, phAKT, phmTOR, and Mcl-1 protein levels as determined by Western blotting in DCs exposed to hypoxia for 24 h and treated or untreated in the last 6 h with Wortmannin. (**B**) phAKT, phmTOR, and Mcl-1 protein levels as determined by Western blotting in DCs exposed to hypoxia for 24 h in the presence or not of LPS. (**C**) RNASET2, mRNA, and protein levels as determined by RT-qPCR analysis and Western blotting, respectively, in DCs exposed to hypoxia in the presence or not in the presence of LPS for 24 h. RNASET2 protein levels as determined by Western blot analysis in DCs exposed to hypoxia in the presence of LPS for 24 h and treated or untreated in the last 6 h with Wortmannin. All blots shown are representative of at least three independent experiments and β-actin was used as loading control. β-actin was used as a housekeeping gene for RT-qPCR analysis. * indicates statistically significant differences (*p* ≤ 0.05; *n* = 3).

## Data Availability

The raw data supporting the conclusions of this article will be made available by the authors, without undue reservation.

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
