# Peer review of "Hypoxia Enhances the Expression of RNASET2 in Human Monocyte-Derived Dendritic Cells: Role of PI3K/AKT Pathway"

_ijms, 2021, doi:10.3390/ijms22147564_

Round 1
Reviewer 1 Report
In the present work, Monaci S. et al showed that hypoxia increased the expression and secretion of the T2 extracellular ribonuclease RNASET2 in human monocyte-derived DCs. This effect was accompanied by an accumulation of HIF-1a and it was negatively regulated by PI3K/AKT patway and LPS. The study represents a continuation of previous investigations by the same authors meaning they appear quite expert in the field.
The experiments are appropriately designed and the data are clearly presented. In the conclusion section, the authors speculate that 'RNASET2 upregulation, together with the inhibition of phAKT, phmTOR and Mcl-1 might be part of an “editing” process that leads to the elimination of pro-tumor immature DCs. This process will abort in the case DCs sense activation signals, such as LPS, leading to maturation and anti-tumor immunity'. In order to state that, it would be interesting to more deeply investigate the relation between RNASET2 expression and the DC maturation profile (expression of CD80, CD86 and MHCII, OVA uptake) in hypoxia condition. This further information would improve the work and render it more complete.
Author Response
Response:
We thank the Reviewer for the positive comments regarding the introduction, the relevant references, the research design, the methods, and the presentation of the results.
We agree with the fact that some of the conclusions may be improved by additional data. However, in this short communication, the first question we wanted to answer was whether hypoxia could induce RNASET2 expression in human dendritic cells and whether PI3k /AKT was involved. The LPS data opened a new perspective, suggesting a correlation between DC maturation levels and RNASET2 expression (and function). It is therefore obvious that some experiments, such as those indicated, are currently underway. Those experiments will be crucial to further study the mechanisms through which RNASET2 could be involved in the so-called editing, promoting the death of immature DCs, which are deleterious in TME, being associated with an inefficient immune response against the tumor. As suggested by Reviewer #2, in the revised manuscript we mention the limitations of our work in a specific paragraph in the discussion.
Reviewer 2 Report
The current manuscript describes a brief communication study on the effect of that hypoxia enhances the expression
and secretion of RNASET2 in human monocyte-derived DCs. This paralleled the HIF-1α accumulation and HIF-dependent and independent signaling, and with a decreased DCs survival. Tellingly, the authors found out that RNASET2 overexpression, under hypoxia, was 82
enhanced by pharmacologically inhibition of PI3K/AKT. Furthermore, treatment of hypoxic DCs with the TLR4 ligand LPS resulted in the abolishment of RNASET2 expression, which was reversed by PI3K/AKT inhibition.
Overall, the current study, albeit preliminary, is of potential interest and could be published after addressing the following points:
1- All the figures of western blots must be displayed in higher resolution and higher quality. in its current form , the blots are unclear and thus should be displayed in better format, probably in better cropped version.
2- Since the current work mainly observational and correlational, it would be important that the authors mention the limitations of their work in separation section in the discussion.
Author Response
- All the figures of western blots must be displayed in higher resolution and higher quality. in its current form , the blots are unclear and thus should be displayed in better format, probably in better cropped version.
Response:
We thank the Reviewer for finding our brief communication, albeit preliminary, of potential interest.
We have earnestly tried to comply with Reviewer’s indications, and in the revised manuscript, all the figures of WB are now displayed in a higher resolution and quality (600dpi).
- Since the current work mainly observational and correlational, it would be important that the authors mention the limitations of their work in separation section in the discussion.
Response:
We agree with the Reviewer, and we added a paragraph in the discussion where we mentioned the limitations of our work. In this “short communication”, the first question we wanted to answer was whether hypoxia could induce RNASET2 expression in human dendritic cells and whether PI3k / Akt was involved. The LPS data opened a new perspective, suggesting a correlation between DC maturation levels and RNASET2 expression and function. All this is currently being studied and, hopefully, it will be the subject of a specific manuscript.
Round 2
Reviewer 1 Report
I thank the authors for their response. I appreciated the integration of the discussion section with a paragraph that declares the limitations of the work and anticipates future experiments focused to exploring the mechanisms that could correlate the modulated expression of RNASET2 and the maturation status of the DCs.
Thanks for the English language improvement.
In the present form, the paper is acceptable for pubblication.